# A Fresh Look at Grape Powdery Mildew (*Erysiphe necator*) A and B Genotypes Revealed Frequent Mixed Infections and Only B Genotypes in Flag Shoot Samples

**DOI:** 10.3390/plants9091156

**Published:** 2020-09-07

**Authors:** Anett Csikós, Márk Z. Németh, Omer Frenkel, Levente Kiss, Kálmán Zoltán Váczy

**Affiliations:** 1Food and Wine Research Institute, Eszterházy Károly University, H-3300 Eger, Hungary; csikosanett1@gmail.com; 2Plant Protection Institute, Centre for Agricultural Research, H-1525 Budapest, Hungary; nemeth.mark@agrar.mta.hu (M.Z.N.); Levente.Kiss@usq.edu.au (L.K.); 3Department of Plant Pathology and Weed Research, Institute of Plant Protection, Agricultural Research Organization (ARO), The Volcani Center, Bet Dagan 50250, Israel; omerf@volcani.agri.gov.il; 4Centre for Crop Health, Institute for Life Sciences and the Environment, University of Southern Queensland, Toowoomba 4350, Australia

**Keywords:** overwintering, population structure, sympatric genetic differentiation, temporal isolation, vineyards, *Vitis vinifera*

## Abstract

*Erysiphe necator* populations, causing powdery mildew of grapes, have a complex genetic structure. Two genotypes, A and B, were identified in most vineyards across the world on the basis of fixed single nucleotide polymorphisms (SNPs) in several DNA regions. It was hypothesized that A populations overwinter as mycelia in grapevine buds, giving rise to so-called flag shoots in spring, and are more sensitive to fungicides than B populations, which overwinter as ascospores and become widespread later in the season. Other studies concluded that the biological significance of these genotypes is unclear. In the spring of 2015, there was a unique opportunity to collect *E. necator* samples from flag shoots in Hungary. The same grapevines were sampled in summer and autumn as well. A total of 182 samples were genotyped on the basis of β-tubulin (*TUB2*), nuclear ribosomal DNA (nrDNA) intergenic spacer (IGS), and internal transcribed spacer (ITS) sequences. Genotypes of 56 samples collected in 2009–2011 were used for comparison. Genotype A was not detected at all in spring, and was present in only 19 samples in total, mixed with genotype B, and sometimes with another frequently found genotype, designated as B2. These results did not support the hypothesis about temporal isolation of the two genotypes and indicated that these are randomly distributed in vineyards.

## 1. Introduction

Sympatric genetic differentiation of plant pathogens that occur primarily or exclusively in agricultural or horticultural fields and infect a single crop species only is still little understood. Apparently, this is the case in populations of *Erysiphe necator* (syn. *Uncinula necator*), the causal agents of grape powdery mildew, one of the economically most important plant diseases in the global crop protection market in terms of fungicide applications [1]. Despite its economic importance, the study of *E. necator* populations is notoriously difficult, due to the obligate biotrophic nature of the pathogen, as well as the rather complex genetic structure of its populations revealed in many regions across the world. Two distinct genetic groups, A and B, have been consistently identified in the Western European and Australian *E. necator* populations, first using DNA fingerprinting methods [2,3], and subsequently on the basis of fixed nucleotide differences, i.e., single nucleotide polymorphisms (SNPs) in sequences of the eburicol 14α-demethylase (*CYP51*) gene [4], the β-tubulin (*TUB2*) gene [5], and also in sequences of the nuclear ribosomal DNA (nrDNA) internal transcribed spacer (ITS) region [4], and nrDNA intergenic spacer (IGS) regions and translation elongation factor 1-α (*EF1-α*) [6].

Most of the population genetic studies carried out thus far have focused on Western European and Australian *E. necator* populations [2,3,4,5,7,8,9,10,11,12,13], with a few additional strains originating from Asia [2,4]. Brewer and Milgroom [6] conducted extensive sampling in the United States and also included some representative strains from Italy and France. Their results suggested that *E. necator* may have been introduced to Europe and then to Australia from the eastern part of the USA in the 19th century, as suggested earlier [14]. However, their study did not rule out the possibility that *E. necator* may have originated from other regions [6]. A major drawback of population genetic studies of *E. necator* is that large areas of its occurrence, e.g., grapevine-growing regions in Central and Eastern Europe, Asia, Australia, and South America, have not been extensively sampled yet. Recently, reports from Chile [15], New Zealand [16], India [17], and Portugal [18] have provided data about the structure of local populations of *E. necator* in these countries, confirming the presence of genotypes A and B in all vineyards studied. The biological significance of the genetic differentiation in *E. necator* populations remains unclear [1], although it has been hypothesized that group A populations are clonal, and group B populations are sexual [2,5,7,8]. Differences in the presumed temporal distribution of these two genetic types has also been described in some vineyards because it was shown that group A isolates tend to disappear during the season, whereas group B isolates were detected during the entire season [8,9,10]. Miazzi et al. [11] argued that the disappearance of group A during the epidemic is due to its higher sensitivity to azole fungicides applied during the season. Furthermore, differences in the overwintering of group A and B have also been suggested. *Erysiphe necator* can overwinter as either resting asexual mycelium in grapevine buds that start sporulating on the emerging young shoots in the spring, causing the so-called flag shoot symptoms [19], or as chasmothecia that remain attached to the bark of grapevines during winter and release ascospores at the beginning of the season [20,21,22,23]. Isolates representing group A were reported to overwinter as resting mycelium in buds, causing flag shoot symptoms, while group B isolates were considered as being released as ascospores from chasmothecia in spring [5,24,25]. Other studies reported that both groups of isolates can lead to flag shoot symptoms [7,12,13]. Therefore, the distinction of these two genetic types based solely on the symptoms observed in spring has been considered as imprecise [1], and more studies are needed to investigate whether groups A and B differ in terms of their overwintering. Differences in the virulence patterns of the two groups have also been described [8,9,10]. It was suggested that genotypes A and B may be reproductively isolated in the field, as some laboratory crosses did not produce viable ascospores [24]; however, further experiments showed that the two groups are, in fact, interfertile, as viable progenies were obtained when monoconidial isolates were mated in the laboratory [25,26].

In the spring of 2015, a unique opportunity arose in Hungarian vineyards to re-visit the controversies regarding the *E. necator* genotypes responsible for flag shoot symptoms and the presumed temporal isolation of genotypes A and B in vineyards during growing seasons. That year, most probably due to the mild winter period, flag shoots emerged in a number of vineyards across the country. Powdery mildew samples were collected and genotyped from some of the flag shoots in spring, as well as from the same grapevine plants in summer and autumn, in order to follow the dynamics of different *E. necator* genotypes during the season. Powdery mildew samples collected in previous years in Hungary were also genotyped in this work on the basis of *TUB2*, IGS, ITS, and *EF1-α* sequences to obtain a more comprehensive picture of the genetic structure of *E. necator* populations in this region. A preliminary analysis based on *TUB2* and *CYP51* sequences of *E. necator* samples collected at the end of a single growing season in Hungary [27] was a useful first step in this respect. The main goal of this work was to take another look at the temporal dynamics and the biological significance of the *E. necator* genotypes in commercial vineyards across the growing season, focusing on powdery mildew infections of grapevines that produced flag shoots at the beginning of the vegetation period.

## 2. Results

### 2.1. Direct Sequencing of the Target TUB2, ITS, and IGS Regions

An approximately 500 bp long fragment of the *TUB2* gene, referred to as the “long *TUB2* fragment” in this work (Figure 1), was PCR-amplified and sequenced at both strands, then analyzed in all the 182 flag shoot samples obtained in 2015, as well as the 32 other *E. necator* samples collected from 2009 to 2011 (Appendix A). On the basis of chromatograms, only 27 flag shoot samples and 11 others represented a single “clean” *TUB2* haplotype, i.e., no double peaks at any nucleotide positions. All but one of these 38 samples had cytosine (C) at nucleotide position no. 79, as defined by Brewer and Milgroom [6]. Therefore, these 37 samples belonged to genotype B, and the other one exhibiting thymine (T) at this position represented genotype A because a T/C SNP at this nucleotide position is the only difference in the *TUB2* gene that distinguishes genotypes A and B according to Amrani and Corio-Costet [5] and subsequent studies [5,8,16]. Other samples, 18 in total, had T/C double peaks in their chromatograms at nucleotide position no. 79; this indicated that both genotypes A and B were present in those samples. The rest of the samples, 158 in total, exhibited C at that nucleotide position, and thus belonged to genotype B; these also had C/T double peaks at position no. 368 (Figure 1). Five out of the 18 samples with T/C double peaks at position no. 79 exhibited C/T double peaks at position no. 368 as well. Thus far, this SNP at position no. 368 has only been reported in a single *E. necator* sample from California, and four more from Italy [6].

Brewer and Milgroom [6] have also identified SNPs at 10 other nucleotide positions (nos. 24, 37, 82, 128, 183, 207, 288, 316, 344, and 356) of the longer *TUB2* fragment. None of our samples exhibited any SNPs at these 10 other nucleotide positions. Therefore, in principle, the Hungarian samples could have represented maximum four genotypes, i.e., four combinations of two nucleobase options (C or T) at two positions (nos. 79 and 368) on the basis of the direct sequencing of the longer *TUB2* fragment. To reveal which of these potential four genotypes did actually occur in our samples, we cloned and sequenced the PCR products of a total of 19 selected samples that exhibited double peaks at both nucleotide positions in their *TUB2* fragments.

Direct sequencing of the ITS PCR products in a subset of 46 samples showed that all but one sample represented genotype B on the basis of the nucleotide exhibited at position 48 of the ITS sequence, as defined by Brewer and Milgroom [6]. The only exception was sample K10/Cs610 (Appendix A), which contained a T/C double peak at this position. This was reported as the only SNP in the ITS region of *E. necator* samples collected in Australia, Germany, India, Israel, and Tunisia [4]. In contrast, Brewer and Milgroom [6] reported five more nucleotide positions (nos. 84, 86, 170, 420, and 462) in the ITS region that were polymorphic in North American samples. No double peaks were detected at any of these five positions in the Hungarian samples. A representative sequence of the ITS region of genotype B and the ITS sequence of sample K10/Cs610 with the T/C SNP at position 48 were deposited in GenBank under accession numbers MT820951 and MT820950, respectively.

Direct sequencing of the IGS fragment (Figure 2) in a subset of 18 samples indicated that all but one belonged to group B on the basis of the nucleotides exhibited at positions nos. 108, 216, and 223, as defined by Brewer and Milgroom [6]. Again, sample K10/Cs610 was the only exception because it contained a C/T double peak at these three positions. Brewer and Milgroom [6] reported SNPs in two more nucleotide positions (nos. 206 and 211) in North American and Australian samples, but these were not polymorphic in any Hungarian samples.

### 2.2. Sequencing Following Cloning of Selected TUB2 and IGS PCR Products

TUB2 amplicons obtained from a total of 19 samples that exhibited T/C double peaks at nucleotide positions no. 79 and/or no. 368 of their TUB2 sequences were cloned, and a total of 116 clones were sequenced (Appendix A). In 16 out of the 19 samples, clone sequences revealed the presence of more than one TUB2 genotype. The presence of genotype A was confirmed in 11 samples, and genotype B in all samples included in the cloning work (Appendix A). Genotypes A and B were identified on the basis of the T/C SNP at nucleotide position no. 79, according to Amrani and Corio-Costet [5], Montarry et al. [8], Brewer and Milgroom [6], Cooper et al. [16], and other studies.

This work has also revealed the existence of another TUB2 genotype, designated as B2 in this study, and characterized by nucleobase C at position no. 79, diagnostic for genotype B, and T at no. 368. Genotype B2 was confirmed by a total of 34 clone sequences obtained from 8 out of the 19 samples included in the cloning work (Appendix A). In two samples, LH09/41 and LH15/S17, sequences of all the three TUB2 genotypes, i.e., A, B, and B2, were confirmed. Representative sequences of the TUB2 genotypes A, B, and B2 were deposited in GenBank under accession numbers MT829251, MT829252, and MT829253, respectively.

On the basis of the chromatograms of the directly sequenced TUB2 PCR products, a fourth genotype, characterized by nucleobase T at both positions nos. 79 and 368, has also been hypothesized. However, such a genotype was not identified in either this work or by Brewer and Milgroom [6], although they revealed a total of 13 TUB2 genotypes in samples collected in the USA, Italy, France, and Australia.

In sample K10/Cs610, chromatograms resulted from direct sequencing of the IGS fragment revealed SNPs at nucleotide positions nos. 108, 216, and 223, indicating that both IGS genotypes A and B are present in the sample. Sequencing following cloning resulted in five clone sequences diagnostic for IGS genotype A, and two for B, thus confirming the results of direct sequencing. A representative sequence of each the two IGS genotypes was deposited in GenBank under accession numbers MT829249 and MT829250.

### 2.3. Genotypes Identified in Flag Shoot Samples

All the 71 flag shoot samples collected in May 2015 in seven vineyards represented genotypes B and/or B2 (Appendix A). Genotype A was present in only 10 samples collected in August and September/October 2015, and genotypes B and/or B2 were also present on the same sampled leaves. Both B and B2 genotypes were widespread in all vineyards during the entire vegetation period in 2015, and were also frequently found, sometimes on the same leaves, in 2009–2011. Genotype A was only found in 9 samples out of 56 in 2009–2011, and was always mixed with B and/or B2 (Appendix A).

### 2.4. Cleaved Amplified Polymorphic Sequence (CAPS) Analysis of the TUB2 and IGS Fragments

To assign the *E. necator* samples to the *TUB2* and IGS genotypes A or B using a less expensive and faster method than sequencing, we carried out CAPS analyses of the long and the short *TUB2* fragments and IGS PCR products according to the method described by Montarry et al. [8] (Appendix A). The CAPS analysis of a total of 211 *TUB2* PCR products identified the same genotypes in the respective samples as direct sequencing and sequencing following cloning, except for 10 samples. In those samples, CAPS analysis did not detect some of the genotypes revealed by the sequencing work (Appendix A).

In case of the IGS PCR products, CAPS analysis identified all but one of the 101 samples as genotype B, similar to the sequencing work. One sample, K10/Cs610, was a mixture of genotypes A and B. This was also revealed by direct sequencing of the IGS fragment in K10/Cs610 (Appendix A). Fragment sizes of the *TUB2* and the IGS PCR products used in CAPS analyses are shown in Table 1.

## 3. Discussion

In regions with severe frost periods during winter, flag shoots are relatively rarely observed in spring [1]. Hungary is characterized by harsh winters, and *E. necator* mainly overwinters as ascospores in chasmothecia attached to the bark of grapevine trunks and branches; flag shoots are very rarely seen in spring [28]. In 2015, following a relatively mild winter period, there was a unique opportunity to detect flag shoots in two wine regions of the country, and test the controversial hypothesis on the genetic differentiation and temporal isolation in *E. necator* populations [1]. A large number of powdery mildew samples were genotyped from and around grapevine plants with flag shoots in spring, summer, and autumn 2015, in both wine regions. These data were analyzed together with the genotypes of other *E. necator* samples collected randomly in vineyards from 2009 to 2011. This second set of samples was needed to reveal which *TUB2*, IGS, and ITS genotypes of *E. necator* were present in Hungarian vineyards before the study period.

In terms of methodology, our work has largely supported the value of the CAPS analysis of both *TUB2* and IGS fragments, similar to a recent study [15], although, as expected, direct sequencing has revealed more SNPs compared to CAPS. Sequencing of PCR products following cloning was applied in this work to validate sequence variations indicated by double peaks in chromatograms of directly sequenced amplicons. Clone sequences from 16 of 19 cloning experiments have supported the presence of SNPs in such cases.

In the USA, Brewer and Milgroom [6] showed that *E. necator* populations are much more diverse, detecting altogether 45 genotypes based on *TUB2*, IGS, ITS, and *EF1-α* sequences. Our data indicated a much lower genetic diversity in the Hungarian *E. necator* populations—all but one sample represented genotype B on the basis of both ITS and IGS sequences, and only three genotypes, A, B, and B2, were identified on the basis of *TUB2* sequences. The *EF1-α* sequences determined in selected Hungarian samples did not reveal any polymorphisms (data not shown). A similar genotyping study carried out in New Zealand vineyards has revealed a much higher genetic diversity, including SNPs in *EF1-α* sequences [16].

In Hungary, most SNPs were detected during *TUB2* sequence analyses. One of these, which defined a genotype designated here as B2, was widespread in all our samples collected from 2009 to 2015. Earlier, this *TUB2* genotype was found only once in California, and four times in Italy [6]. As the genetic structure of grape powdery mildew populations has not been intensively studied yet in Central Europe, the overall distribution of this *TUB2* genotype remains unclear. Moreover, some of the earlier genotyping studies did not amplify and examine the region that contains position 368 of the *TUB2* sequence [8]; thus, genotype B2 may be more widespread than it appears.

The grapevine powdery mildew samples genotyped in this work came directly from the field, while several other studies carried out similar genotyping works using monoconidial *E. necator* isolates produced on detached leaves in the laboratory [5,9,11]. Most of our field samples, each collected from a single leaf, bunch, or shoot starting from 2009 to 2015, were mixtures of two *TUB2* genotypes, B and B2. Only a low number of flag shoot samples from 2015, and a few others collected randomly from 2009 to 2011, represented a single *TUB2* genotype. Monoconidial isolates examined in other works have always represented single genotypes for all loci [5,9,11], including the highly variable microsatellite loci [29]. This indicates that there are no SNPs in *TUB2*, IGS, and ITS sequences of *E. necator* colonies originating from single conidia, although at least the IGS and the ITS regions are present in multiple copies in the haploid ascomycete genomes [30]. Similarly, our blastn search in the contigs of the *E. necator* genome determined by Jones et al. [31] did not reveal divergent *TUB2* genotypes. Therefore, our field samples that yielded at least two *TUB2* genotypes may have consisted of powdery mildew colonies originating from genetically distinct conidia.

Genotype A was only rarely detected in our samples collected randomly from 2009 to 2011. In 2015, when our samples were collected from the same grapevine plants across the growing season, this genotype was again much less frequent than B and B2. A preliminary analysis of *E. necator* populations in Hungarian vineyards revealed the same genetic structure on the basis of samples collected at the end of the season in a single year [27]. Genotype A was also rarely found in Chile [15] and New Zealand [16]. In our study, none of the 182 flag shoot samples collected in spring represented this genotype; all the samples that contained genotype A, usually mixed with B and/or B2, were collected in summer or autumn in 2015.

In conclusion, our study did not support the hypotheses developed around temporal isolation of genotypes A and B during the season, as well as differences in their overwintering, suggested by some authors [4,5,8,9,10,24,25], but questioned by others [1,7,12]. Instead, our data indicated that genotype A can be detected in vineyards later the season, and it is not associated with flag shoot symptoms. The presence of genotype A in the sampled vineyards in summer and/or autumn in 2009, 2010, 2011, and 2015, may also question its higher sensitivity to azole fungicides, presumed by Miazzi et al. [11], because all those vineyards were treated with such fungicides during each season (data not shown). Although fixed SNPs in the *TUB2*, IGS, ITS, and in some regions also in *EF1-α* sequences of *E. necator* clearly define distinct genotypes of this pathogen, the biological significance of their overall distributions in vineyards remains unclear. Further, well-structured and comprehensive genotyping studies covering the entire growing season are needed to reveal whether there are clear patterns that characterize the temporal and/or spatial distribution of genotypes A and B during the season, or these two groups are, in fact, randomly distributed in vineyards.

## 4. Materials and Methods

### 4.1. Erysiphe necator Sample Collection from 2009 to 2011

To determine which *TUB2*, IGS, and ITS genotypes of *E. necator* are present in Hungarian vineyards, we collected a total of 56 powdery mildew-infected grape samples from 2009 to 2011 from across the country (Appendix A). All these samples were included in the cleaved amplified polymorphic sequence (CAPS) analysis, and 31 out of 56 also in the sequencing work. Each sample consisted of a single leaf or a berry, partially or completely covered with powdery mildew mycelium. Upon collection, each plant sample was placed in a new paper bag and transported to the laboratory in a cool box to be processed in less than 24 h. To avoid cross-contamination amongst samples collected on the same day, or in the same place, we collected all of the samples at least 300 m apart from each other, and hand sanitizer was applied after each sample collection. In the laboratory, a part of the fresh powdery mildew material was removed from each sample with a sterile artist’s brush, and placed in a sterile 1.5 mL tube, in isolation from any other *E. necator* material. After closing the tubes, and before labelling, tubes were always cleaned with 70% ethanol to remove any powdery mildew material that may have remained on their external surfaces. Total genomic DNA was extracted from powdery mildew materials using a Qiagen DNeasy Plant Mini Kit (Qiagen GmbH, Germany) or Macherey-Nagel NucleoSpin-Plant II Kit (Macherey-Nagel GmbH, Germany) according to the protocols provided by the manufacturers. DNA samples were stored at −18 °C until use.

### 4.2. Erysiphe necator Samples Collected from Flag Shoot Sites in 2015

In spring 2015, an unusually high prevalence of flag shoots was observed in vineyards of two Hungarian wine regions, Eger and Szekszárd, located approximately 300 km apart from each other. To determine the *E. necator* genotypes that are present on flag shoots in spring, and then on the same grapevine plants later in the season, we collected a total of 182 samples altogether in seven vineyards of the two wine regions from *V. vinifera* cultivars Kékfrankos and Blauburger. The first sampling was performed in May, the second in August, and finally the third at the end of the growing season in late September and early October 2015 (Appendix A). The sampling method was as described above. In May, powdery mildew-infected leaf samples were collected directly from the flag shoots, which were marked with colored adhesive tapes to be identified later in the season. In August, and then in September–October, infected leaf and berry samples were collected randomly from the same plant or from grapevine plants situated next to the one with flag shoot symptoms in spring. All the 183 samples collected in 2015 are referred to as “flag shoot samples” in this work.

### 4.3. PCR Amplification of the Target TUB2, ITS, and IGS Regions in E. necator Samples

The “long *TUB2* fragment” (Figure 1) was amplified using primers Bt2c and Bt2d, as well as the corresponding PCR protocol developed by Brewer and Milgroom [6]. When these amplifications did not yield good quality amplicons and/or chromatograms, we amplified an approximately 200 bp long fragment of *TUB2*, referred to as the “short *TUB2* fragment” in this work (Figure 1), with new primers, TUB2_Fw (5′-AATTGGGGCTGCTTTCTGTA-3′) and TUB2_Rev (5′-AACATTCATCCTTTCGAGCTG-3′) developed in this work on the basis of a published full-length coding sequence of *TUB2* in *E. necator* (GenBank accession no. AY074934 [6]). PCRs were carried out as described by Brewer and Milgroom [6] for *TUB2* amplifications. The primer combinations TUB2_Fw/Bt2d and Bt2c/TUB2_Rev were also used on the basis of this protocol to amplify the long *TUB2* fragment in two steps.

Two regions of the nuclear ribosomal DNA (nrDNA) have also been amplified and analyzed in this work. The ITS region of nrDNA was amplified with primers ITSEnF and ITSEnR, as described earlier [6]. An approximately 300 bp long fragment of the IGS region of nrDNA (Figure 2) was amplified with primers IGS_Fw (5′-GTTGGGATCCTCCTCCAG-3′), developed in this work on the basis of the IGS sequence (GenBank accession no. GQ255476 [6]), and NS1R [32]. PCRs were performed as described by Frenkel et al. [29] for microsatellite amplifications.

All PCRs were carried out using either BIO-X-ACT Short Mix (Bioline, Alvinston, ON, Canada) or DreamTaq Green PCR Master Mix (Thermo Fisher Scientific, Waltham, MA, USA) as recommended by the manufacturers. A negative control lacking template DNA was included in each set of PCRs. PCR products were verified using DNA High Resolution Gel Cartridges with the QIAxcel Instrument, and visualized by QIAxcel ScreenGel software (Qiagen, Hilden, Germany).

### 4.4. Direct Sequencing of the Target TUB2, ITS, and IGS Regions

PCR products were purified with Quantum Prep PCR Kleen Spin Columns (Bio-Rad, Hercules, CA, USA) or an EZ-10 Spin Column PCR Products Purification Kit (Bio Basic Inc., Markham, ON, Canada), and sent for direct sequencing of both strands to LGC Genomics GmbH (Berlin, Germany). Sequencing was conducted with the primers used for the respective amplifications.

### 4.5. Sequencing Following Cloning of Selected TUB2 PCR Products

*TUB2* PCR products, obtained as described above from 17 samples, and showing single nucleotide polymorphisms (SNPs) in their chromatograms, were cloned into the pJET1.2/blunt vector to determine whether these are mixtures of two or three different *TUB2* genotypes. The IGS PCR product obtained from a single sample, K10/Cs610, was also cloned in the same way because its chromatograms showed double peaks at the nucleotide positions that distinguish genotypes A and B. Cloning was conducted using the CloneJET PCR Cloning Kit (Thermo Fisher Scientific) according to the manufacturer’s protocol. Competent *Escherichia coli* cells were prepared as described by Inoue et al. [33] and transformed with heat shock at 42 °C for 1 min. Suspensions were spread onto lysis broth medium containing 100 mg/mL carbenicillin (Duchefa Biochemie). Positive clones were selected by removing a part of the transformed bacterial colonies with sterile toothpicks and adding bacteria directly to the PCR mixtures. For these amplifications, primers provided with the kit were used, and the recommended PCR protocol was followed. Up to 10 clones per sample were sequenced with pJET1.2 Forward and reverse sequencing primers were from LGC Genomics GmbH. PCR products resulting from colony-based amplifications of seven samples were also included in CAPS analyses with *Xmi*I, as described below.

### 4.6. Sequence Analyses

All chromatograms resulting from direct sequencing and sequencing following cloning were visually checked for the presence of double peaks indicating SNPs as detailed by Lesemann et al. [34] and Kovács et al. [35]. Sequences were compiled from electrophoregrams using Pregap4 and Gap4 [36] and were aligned using Clustal Omega [37] or MAFFT Online version 7 [38]. Variable nucleotide positions in alignments were visualized with MEGA7 [39]. Representative sequences of all *TUB2*, ITS, and IGS genotypes were deposited in National Center for Biotechnology Information (NCBI) GenBank. Primer map illustrations (Figure 1 and Figure 2) were prepared using SnapGene Viewer software (version 4; GSL Biotech, San Diego, CA, USA).

To verify whether there are two divergent *TUB2* copies in the *E. necator* genome, as presumed in another powdery mildew species [40], we conducted a blastn search against sequenced *E. necator* genomic contigs [31], with the *E. necator TUB2* reference sequence GQ255475 [6] as a query. Significant hits (E score = 0.0; 100% cover) contained only identical *TUB2* sequences from the publicly available genomes, and thus no other closely related regions were revealed in the *E. necator* genome.

### 4.7. Cleaved Amplified Polymorphic Sequence (CAPS) Analysis of the TUB2 and IGS Fragments

To assign the *E. necator* samples to the *TUB2* genotypes A or B, we conducted a CAPS analysis of the long and the short *TUB2* fragments, as described by Montarry et al. [8]. Their CAPS method is based on the detection of a T/C SNP between A and B genotypes using the restriction endonuclease *Acc*I or its isoschizomer *Xmi*I. The recognition site of these enzymes is GTMKAC (M = A/C, K = G/T). The amplified *TUB2* fragments containing the SNP specific to *TUB2* genotype B, thus the recognition site GTAGAC, were digested, while the fragments with the recognition site GTAGAT that lack this SNP remained undigested; this is diagnostic for genotype A (Table 1).

To conduct CAPS analyses of the long and short *TUB2* PCR products, these were digested with the enzyme *Xmi*I (Thermo Fisher Scientific) in 20 µL final volume. Digestion reaction mixes included 12.5 µL of the PCR product, 2 µL Buffer B from the kit, 0.5 µL restriction enzyme (10 U/µL), and 5 μL nuclease-free molecular biology water (Thermo Fisher Scientific). Digestion was performed at 37 °C for 2.5 h. Fragment analysis following the digestion reaction was conducted with Screening Gel Cartridge in the QIAxcel Instrument and visualized by QIAxcel ScreenGel software (Qiagen, Germantown, MD, USA) (Appendix A).

The CAPS analysis of the PCR-amplified IGS fragment was conducted with restriction endonuclease *Xho*I (Thermo Fisher Scientific), and the same reaction composition and subsequent fragment analysis as described above for *TUB2* CAPS. This method identifies genotypes A and B on the basis of a C/T SNP at nucleotide no. 108 of the reference IGS sequence GQ255476 [6]. During the digestion reaction, the IGS fragments diagnostic for genotype A are digested, while those diagnostic for genotype B remain unchanged (Table 1).

## Figures and Tables

**Figure 1 plants-09-01156-f001:**
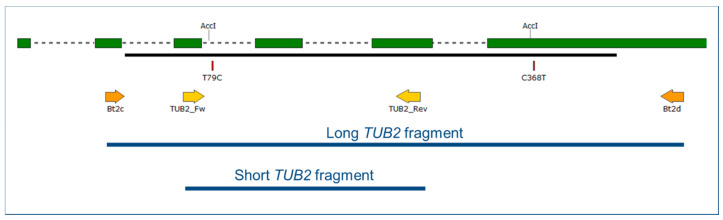
Map of the 5′ region of the *TUB2* gene of *Erysiphe necator*. Green boxes illustrate coding regions in the first ≈600 base pairs of the gene starting from the ATG start triplet; dashed lines in between boxes indicate introns. Black line shows region covered by GQ255475 sequence used as reference. Arrows denote primers used in this work. T79C and C368T indicate positions of detected single nucleotide polymorphisms differentiating A from B and B2 genotypes, and B2 from A and B, respectively. Positions labeled with AccI indicate *Acc*I restriction enzyme sites used in the Cleaved Amplified Polymorphic Sequence (CAPS) analysis. The long and the short *TUB2* fragments are shown in blue.

**Figure 2 plants-09-01156-f002:**
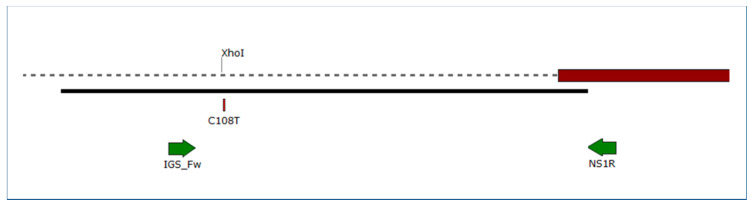
Map of the 3′ region of the nuclear ribosomal DNA (nrDNA) intergenic spacer (IGS), and the 5′ end of the ribosomal large subunit (28S) gene in *Erysiphe necator*. Dashed line illustrates IGS and red box indicates 5′ end of the 28S gene. Black line shows the region covered by GQ255476 sequence used as reference. Arrows denote primers used in this work. C108T indicates the position of the single nucleotide polymorphism differentiating A and B genotypes. The position labeled with *Xho*I shows the *Xho*I restriction site used in our Cleaved Amplified Polymorphic Sequence (CAPS) analysis, present in genotype A, but not in B.

**Table 1 plants-09-01156-t001:** Summary of Cleaved Amplified Polymorphic Sequence (CAPS) analyses of the beta-tubulin (TUB2) and the nrDNA intergenic spacer (IGS) fragments.

Target	GenBank Acc. No. of the Reference Sequence	Primer Sequence (5′ → 3′)	Amplicon Size (bp)	Restriction Enzyme	Enzyme Recognition Site	SNP Position	Detected SNPs	Geno-Type	Expected Sizes (bp)
short *TUB2* fragment	GQ255475	TUB2_Fw: AATTGGGGCTGCTTTCTGTA	213	*Acc*I	GTMKAC	79	T	A	213 (not digested)
		TUB2_Rev: AACATTCATCCTTTCGAGCTG					C	B	190, 23
long *TUB2* fragment		Bt2c: CAGACTGGCCAATGCGTA	520			79, 368	T, C	A	32, 138
		Bt2d: AGTTCAGCACCCTCGGTGTA					C, C	B	289, 138, 93
							C, T	B2	427, 93
IGS fragment	GQ255476	IGS_Fw: GTTGGGATCCTCCTCCAG	295	*Xho*I	CTCGAG	108	C	A	260, 35
		NS1R: GAGACAAGCATATGACTAC					T	B	295 (not digested)

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
