# Peer review of "A Fresh Look at Grape Powdery Mildew (Erysiphe necator) A and B Genotypes Revealed Frequent Mixed Infections and Only B Genotypes in Flag Shoot Samples"

_plants, 2020, doi:10.3390/plants9091156_

Round 1

Reviewer 1 Report

Dear Editor, I am sending you my review of the manuscript entitled " fresh look at grape powdery mildew (Erysiphe necator) TUB2, ITS and IGS genotypes in flag shoot samples: Frequent mixed infections and high prevalence of a little known TUB2 genotype” In this manuscript, the authors evaluate genetic structure of E. necator, the casual agents of grape powdery mildew. The study is based on PCR sequencing and CAPS analysis. However, the authors don´t show any CAPS results (agarose gel) or phylogenetic tree to clarified their results The text is also a bit unclear in the discussion. I suggest introduce any gel photograph or even better a phylogenetic tree which could be the text more understandable.

Author Response

Point 1: The authors don't show any CAPS results (agarose gel) or phylogenetic tree to clarified their results

Response 1: As stated in the Materials & Methods (line 369 of the original submission) CAPS analyses were done with QIAxcel, instead of agarose gels. This tool makes fragment analysis easy, elegant, and less time-consuming than the gel-based methods. We included a supplementary figure (Suppl. Fig. 1) in the revised manuscript to provide an example for a QIAxcel image of our CAPS analysis.

Point 2: The text is also a bit unclear in the discussion. I suggest introduce any gel photograph or even better a phylogenetic tree which could be the text more understandable.

Response 2: Both Reviewers 2 & 3 considered that the manuscript is generally well-written. Some parts of the text, including the Discussion, were revised based on specific comments received from Reviewers 2 & 3. To show all the changes made in the original submission during revision, we uploaded a file that highlights these with 'track changes'.

Instead of gels, we used the QIAxcel instrument for fragment analyses. This was stated in the Materials & Methods (line 369 of the original submission). During revision, a supplementary figure was added to the manuscript to provide an example for a QIAxcel image of our CAPS analysis.

The manuscript does not contain any data that could be analyzed from a phylogenetic point of view.

Reviewer 2 Report

This manuscript reports on the molecular genotyping of Erysiphe necator isolates collected from flag shoots in Hungary based of sequences of TUB2, IGS and ITS. Data from this study invalidate the previous hypothesis about seasonal prevalence of the 2 main genotypes A and B in E. necator populations and denote a random distribution of those genotypes in vineyards. Moreover, authors have provided data supporting the presence of a new genotype designated as B2 in some of those isolates.

Overall, the manuscript is very well organized and written. It contains interesting findings that might be of great interest for Plants readers, particularly research groups working on E. necator infections of grapevines. Some minor revisions are requested before acceptance for publication in Plants. Here are some specific edits that should be done to improve the current version of the manuscript:

Minor revisions:

  • The introduction is well-written and provide a comprehensive overview of the previous genetic studies carried out to study the distribution and structure of necator populations worldwide. However, the introduction does not emphasize on the significance of the current study and the importance of genotyping and differentiation between the genetic groups of E. necator. I suggest adding a small paragraph to state the importance of such analyses.

  • Line 183-184: In Table 1, add the unit “bp” in the last column for expected sizes.

  • Throughout the manuscript replace “at random” with “randomly”.

  • Line 236: The samples used for comparison with the 2015 results were collected from years 2009-2011 or 2009-2014? The years of sampling were sometimes stated wrong in the text. Please verify.

  • The manuscript is missing a conclusion paragraph to highlight again the importance of this study in the field and state some future directions.

Author Response

Point 1: The introduction does not emphasize on the significance of the current study and the importance of genotyping and differentiation between the genetic groups of E. necator. I suggest adding a small paragraph to state the importance of such analyses.

Response 1: Thank you for this very helpful suggestion. The last paragraph of the Introduction was completely re-written in line with this comment. We acknowledge that this change improved the manuscript considerably. To show all the changes made in the original submission during revision, we uploaded a file that highlights these with 'track changes'.

Point 2: In Table 1, add the unit “bp” in the last column for expected sizes.

Response 2: Done. Please note that Table 1 is included in the main manuscript file as an image, and also uploaded separately as an .xlsx file.

Point 3: Throughout the manuscript replace “at random” with “randomly”.

Response 3: Done.

Point 4: Line 236: The samples used for comparison with the 2015 results were collected from years 2009-2011 or 2009-2014? The years of sampling were sometimes stated wrong in the text. Please verify.

Response 4: These samples were collected in 2009-2011. There was a typo in line 236 of the original submission, which was corrected during revision. We scanned the entire manuscript and did not find this error elsewhere.

Point 5:The manuscript is missing a conclusion paragraph to highlight again the importance of this study in the field and state some future directions.

Response 5: Agreed. The last paragraph of the Discussion was revised in line with this comment.

Reviewer 3 Report

Manuscript ID plants-917735 and entitled "A fresh look at grape powdery mildew (Erysiphe necator) TUB2, ITS and IGS genotypes in flag shoot samples: Frequent mixed infections and high prevalence of a little known TUB2 genotype " is interesting and refers to the interesting phytophages such as grape powdery mildew, which are very often overlooked by phytopathologists. Therefore, I am very pleased to review this research.

Overall, the paper is well written and the research is of a good standard considering that it is a “communication” and not a “regular article”. Nevertheless, I have some comments on the current version of the manuscript:

1) the title is too long, please propose a shorter and simpler version according to the rule "one simple message"

2) Tab. 1 – poor format, i.e. words are "cut". Please correct.

3) In my opinion, some data in the supplementary tables is important. I ask the authors to consider whether some data may be included in the table in the main body of the manuscript.

Author Response

Point 1: The title is too long, please propose a shorter and simpler version according to the rule "one simple message"

Response 1: Thank you for this very useful comment. The title was changed to the following: 'A fresh look at grape powdery mildew (Erysiphe necator) A and B genotypes revealed frequent mixed infections and only B genotypes in flag shoot samples'. To highlight all the changes made in the original submission during revision, we uploaded a file that shows these with 'track changes'.

Point 2: Tab. 1 – poor format, i.e. words are "cut". Please correct.

Response 2: Table 2 was prepared in line with the journal's standards. All words are written in full, and also abbreviated (in parentheses) when applicable. "bp" was added to the last column during revision, in line with the 2nd Reviewer's comment. Please note that Table 1 is included in the main manuscript file as an image, and also uploaded separately as an .xlsx file.

Point 3: In my opinion, some data in the supplementary tables is important. I ask the authors to consider whether some data may be included in the table in the main body of the manuscript.

Response 3: We agree with this comment; however, all the important data from Suppl. Tables 1 & 2, such as total number of samples collected in 2009-2011, and then in 2015; number of vineyards sampled; number of samples included in the cloning work, etc. are mentioned in the text, in the Results, Discussion, and also in the Materials & Methods. Therefore, we don't think that another table is needed to show the data that are probably better placed in the text.